# Resistance trend in bacteria isolated from corneal ulcers: A retrospective analysis from Pakistan

Naila Obaid[1], Ayesha Saeed[2], Sidra Abbas[3]*, Shaghufta Perveen[3], Hajira Younas[4]

1 Ophthalmology Department, Fauji Foundation Hospital, Foundation University Islamabad, Rawalpindi, Pakistan, 2 Clinical and Biomedical Research Center (CBRC), Foundation University School of Health Sciences (FUSH), Foundation University Islamabad, Rawalpindi, Pakistan, 3 Microbiology and Biotechnology Research Laboratory, Department of Biotechnology, Fatima Jinnah Women University, Rawalpindi, Pakistan, 4 Pathology Department (Microbiology), Fauji Foundation Hospital, Foundation University Islamabad, Rawalpindi, Pakistan

* sidraas@gmail.com

## Abstract

### Objective

The purpose of this study is to isolate and characterize bacteria from corneal ulcers and screen them for multiple antibiotic resistance, contributing to understanding patterns of resistance and identifying effective treatment strategies.

### Design

A retrospective study was conducted between January 2024 to August 2024

### Subjects, participants and/or controls

The study involved the isolation of pathogenic bacterial strains from 15 corneal ulcer samples.

### Methods, intervention, or testing

We isolated and characterized bacterial strains from corneal ulcer samples, which were routinely collected at an ophthalmology clinic for suspected infectious corneal ulcers and examined their resistance to twenty-six routinely prescribed antibiotics. The bacterial species included methicillin-resistant *Staphylococcus aureus* (MRSA), *Pseudomonas aeruginosa* and *E. coli* among others. Antibiotic resistance patterns were assessed, with a focus on commonly used antibiotics such as ciprofloxacin, gentamycin, and vancomycin.

**Data availability statement:** All relevant data are within the manuscript and its Supporting information files.

**Funding:** The author(s) received no specific funding for this work.

**Competing interests:** NO authors have competing interests.

## Main outcome measures

The study measured the healing rates of corneal ulcer patients, antibiotic resistance levels across bacterial strains, and specific treatment responses to various antibiotics.

## Results

Thirteen out of fifteen corneal ulcer patients healed completely, while two patients developed full corneal opacity due to infection with *P. aeruginosa*. Significant differences in resistance were observed among the bacterial strains. MRSA exhibited the highest resistance levels, particularly to multiple antibiotics. Ciprofloxacin and gentamycin were less effective against the isolated strains, while vancomycin showed reduced resistance, against Gram-positive bacteria. Both ciprofloxacin and co-trimoxazole exhibited strong connections with multiple bacterial strains, indicating high resistance.

## Conclusion

This study underscores the need for ongoing surveillance of antibiotic resistance patterns to guide treatment approaches and slow the spread of resistant bacteria. It also highlights the importance of developing new antibiotics and alternative therapies, with an emphasis on understanding the molecular mechanisms behind resistance to combat the global health posed by antibiotic-resistant infections.

## Introduction

Corneal ulcers are disruption of corneal epithelium, often resulting from collagenases produced by regenerating epithelial cells and inflammatory cells, which contribute to ulcer formation. Once healed, corneal ulcers may lead to corneal scarring, a significant cause of blindness [1]. Epidemiological data indicates that corneal ulcers are primarily caused by trauma, often followed by corneal infections. Men are affected more frequently than women, with the majority of cases occurring in individuals between the age of 21 and 40, typically within the productive working age. Farming is the most common occupation associated with these cases [2,3]. Corneal ulcers can result from bacterial, fungal and viral infections as well as non-infectious causes such as chemical burns, radiation, and extreme temperatures. The most common bacterial pathogens responsible for corneal ulcers include *Staphylococcus aureus, Staphylococcus epidermidis, Streptococcus pneumoniae, Streptococcus pyogenes, Moraxella spp, Pseudomonas aeruginosa, Proteus spp, Klebsiella pneumoniae,* and *Escherichia coli* [4,5]. The most common fungi responsible for corneal infection include *Culvularia* sp, *Aspergillus* sp, *Paecilomyces* sp, *Cladosporium spp, Bipolaris* sp, *Altenaria spp,* and *Candida spp* [6,7]. Among these, *Aspergillus* species are predominant in cases associated with trauma, while *Candida* infections are more frequently seen in individuals with predisposing

factors such as contact lens use, ocular surface diseases and immunosuppression [8,9]. Fungal infections often present with slow-progression ulcers with feathery edges, satellite lesions, and hypopyon formation, making early diagnosis and appropriate antifungal therapy crucial to prevent complications [10,11]. Corneal infections can also be caused by viruses such as *Herpes Simplex, Papillomavirus, Retrovirus, Rubella virus,* and *Adenoviruses* [12], or parasites like *Acanthamoeba* [13]. Non-infectious causes of corneal ulcers include trauma from acidic or alkaline substances, radiation or extreme temperature.

The empirical use of antibiotics is common in ophthalmic infections in Pakistan, often due to limited access to microbiological diagnostic and self-medication practices [14]. Fluoroquinolones, particularly ciprofloxacin are widely prescribed for bacterial keratitis, while fortified aminoglycosides and cephalosporins are used for severe infections [15]. However, inappropriate antibiotic use, including over-the-counter availability, prolonged dosing and the use of broad-spectrum antibiotics without culture-based selection, has contributed to antibiotic resistance [16]. These factors drive the selection of resistant bacterial strains, complicating treatment outcomes and increasing the risk of recurrent or refractory infection. Over the past decade, studies conducted in Pakistan's Sindh region particularly in Hyderabad and Karachi, have identified the highest level (median resistance – 90.5%) of resistance of *Salmonella* species to ciprofloxacin [17]. In Punjab, a recent study demonstrated highest fluoroquinolone resistance in *Staphylococcus aureus*, with prevalence rate of 91% to norfloxacin, 83.9% to ciprofloxacin, 78% to azithromycin, 77% to erythromycin and 69.8% to moxifloxacin [18]. Similarly, Gondal *et al*. [19] reported a surge in carbapenem-resistance Enterobacterales in a Lahore tertiary care hospital, exceeding earlier recorded levels, i.e., 21.84% in another tertiary care hospital [20] and 25.55% Rawalpindi's community settings [21].

Early diagnosis and timely administration of appropriate antimicrobial therapy is crucial for preventing permanent visual loss and enhancing patient's quality of life [22,23]. The choice of antimicrobial treatment is guided by the results of corneal smear, culture and antibiotic sensitivity. Some infections in corneal ulcers are caused by resistant bacteria. Fluroquinolone resistance is particularly observed in cases with prior antibiotic use. *P. aeruginosa* has been reported to exhibit resistance to moxifloxacin [24,25] as well as other fluoroquinolones [26].

This study aims to investigate the etiology and antibiotic sensitivity of pathogens causing corneal ulcers at Fauji Foundation Hospital. The findings will contribute to improving the diagnosis and treatment of this potentially blinding condition, while also addressing the growing issue of microbial antibiotic resistance.

## Materials and methods

### Ethical approval statement

The study obtained ethical approval from the Ethical Committee of the Fauji Foundation Hospital.
Ethics reference number: 804/RC/FFH/RWP.

### Study design

This retrospective study was conducted at the Ophthalmology and Microbiology department of Fauji Foundation Hospital, Rawalpindi, Pakistan Rawalpindi from January 2024 to August 2024. The inclusion criteria comprised all adult patients (age ≥18 years) clinically diagnosed with infectious bacterial who underwent corneal scraping for microbiological analysis during the study period.

Exclusion criteria were: (1) patients with non-infectious keratitis (e.g., herpetic keratitis, neurotrophic keratitis with reduced corneal sensitivity); (2) those with concomitant ocular comorbidities that could confound healing assessment (uncontrolled glaucoma, uveitis, or non-healing ulcers from other etiologies); and (3) systemically immunocompromised patients (uncontrolled diabetes [HbA1c >8%], active malignancy, HIV/AIDS, or chronic immunosuppressive therapy). These criteria ensured homogeneous evaluation of antimicrobial resistance patterns in typical infectious keratitis cases while excluding confounding factors.

## Sample collection, isolation, and identification of strains

In accordance with the Helsinki Declaration of 2013, samples were collected as part of the routine practice of ophthalmology clinic for suspected infectious corneal ulcers [27]. A comprehensive external and slit-lamp biomicroscopic examination was conducted on all suspected corneal ulcers by an ophthalmology resident [28].

Corneal scraping was performed under slit-lamp magnification, after administrating a drop of 0.5% proparacaine hydrochloride (Alcaine, Alcon laboratories) for topical anesthesia. Under sterile conditions, a sterile No. 15 Bard Parker blade was used to collect samples from the leading edge and base of the ulcer. The material obtained was then inoculated onto blood agar and Pronadisa medium (Pronadisa Lab. Conda, Madrid, Spain), chocolate agar (Pronadisa Lab. Conda, Madrid, Spain), MacConkey agar (Pronadisa Lab. Conda, Madrid, Spain), and Sabouraud dextrose agar (Pronadisa Lab. Conda, Madrid, Spain) for culture. The procedures followed the guidelines of the American Society for Microbiology [29]. A smear was also prepared for Gram staining to identify bacterial and fungal morphology.

## Antibiotic susceptibility testing

The disc diffusion method, as described by Bauer *et al*. [30], was employed by placing antibiotic-impregnated discs onto Mueller-Hinton agar plates inoculated with the test bacteria. For cultured isolates, antibiotic susceptibility testing was conducted using Mast Discs (Mast Group, Merseyside, UK). The antimicrobial susceptibility patterns were interpreted using by CLSI M45-A3 (2015), (CLSI, 2015) and the 2022 CLSI M100 standard (CLSI, 2022) by measuring inhibition zone diameters (to the nearest 0.1 mm using calibrated calipers) and comparing them to CLSI breakpoints. For vancomycin and polymyxin B, minimum inhibitory concentration (MIC) testing was performed via broth microdilution (BMD) per CLSI guidelines. Susceptibility to the antibiotics listed in S1 Table was determined.

An antibiotic disc was selected based on Gram staining results and the distinct characteristics cultured colony. The microbiological report provided information on the sensitivity or resistance of each sample to the chosen antibiotic.

## Statistical analysis

Microsoft office excel 2016v was used for the descriptive analysis of antibiotic resistance data. The significance of the data was further validated through a Two Way Analysis of Variance (ANOVA), with and without replication, where a p-value of less than 0.05 was considered significant. Other statistical analysis included estimating the frequency and percentage of interaction among selected antibiotics and bacterial strains. The clades were established based on the similarity of biochemical tests (S2 Table). Strains exhibiting similar results were grouped together using R software version 4.4.1.

## Results and discussion

### Sampling site

The Fauji Foundation Hospital (FFH), which is situated in Punjab, Pakistan, provided the study's sample. The attached map shows the sampling site's geographic context and gives a clear visual depiction of its location within Pakistan's larger geography (Fig 1). Pakistan's province borders, including those of Baluchistan, Sindh, Punjab, Khyber Pakhtunkhwa, Gilgit Baltistan, and Azad Kashmir, are depicted on the main map. A closer look at the sampling site is provided by the expanded depiction of the Punjab province in the inset map. The inset map shows the location of the sampling site, which is Fauji Foundation Hospital (FFH). The map indicates that this hospital is situated in the city of Rawalpindi. Grid coordinates are included in the inset map to aid in precisely locating FFH inside Punjab.

The collection of the sample from FFH in Punjab is significant due to several reasons. Punjab is one of the most populous and agriculturally important provinces in Pakistan [31]. The data collected from this region can provide valuable insights into local environmental and health conditions [32]. Fauji Foundation Hospital is a well-established medical and research facility, ensuring the reliability and accuracy of the collected samples. The hospital's infrastructure supports

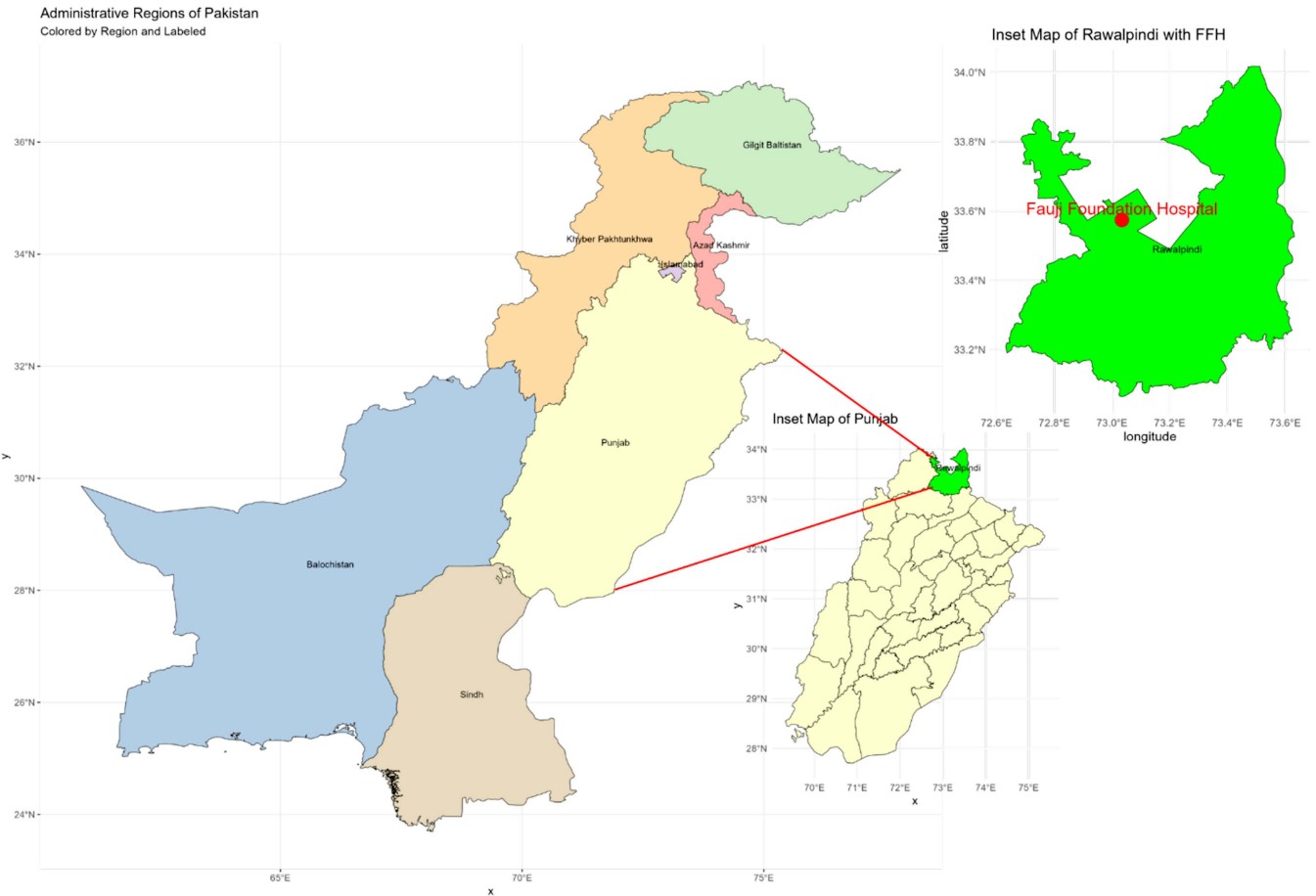

**Fig 1. Map of the study area was generated using geographic coordinates obtained from the Humanitarian Data Exchange (HDX) (** https://data. humdata.org/dataset/cod-ab-pak**).** The map was created in R using open-source mapping packages, ensuring accurate spatial representation of the selected regions.

advanced medical research, making it an ideal location for such studies. The hospital's central location within the province ensures that the sample is representative of the region's demographic and environmental conditions.

## Patient history and sample collection

Samples were obtained from 15 patients aged 10 and 69 years, during the period from January 2024 to August 2024, from the female eye ward at Fauji Foundation Hospital. These samples were collected from suspected infectious corneal ulcers as part of the routine practice in the ophthalmology clinic.

## Patient management and treatment protocol

Each corneal ulcer follows a specific pattern, and treatment in initiated accordingly at the time of presentation, even before the Gram-staining report. Upon presentation, corneal ulcer scrapings were collected and sent for Gram-staining, culture and sensitivity testing. However, treatment of all the patients began immediately after they were diagnosed with corneal ulcers. If the culture and sensitivity report later indicate a change is necessary, treatment can be adjusted accordingly. All the patients were admitted for 5–7 days in the eye ward of the hospital. In recent years, there has been more emphasis on culture and sensitivity testing, as the number of resistant corneal ulcers cases has risen.

### Isolation of strains and morphological characteristics

The colony morphology of the bacterial isolates was assessed based on their appearance. All colonies were circular with raised elevation and smooth margins. Of the fifteen isolates, five were cocci, and the remaining were classified as bacilli. All the isolates were mesophiles, exhibiting growth at pH 7–9. Isolates EW8, EW9, EW11, EW12, EW13, EW15 had mucilaginous, while other displayed a creamy texture.

### Pathogen-specific responses in corneal ulcer treatment

Out of the 15 patients, 13 experienced complete healing of the infection following treatment. However, two patients developed full corneal opacity, which occurred because they sought medical attention almost one week after the onset of the ulcer. In both cases, the causative agent was *P. aeruginosa*. The corneal ulcer (as shown in Fig 2a) was taken after two weeks of treatment and shows the ulcer fully healed. The ulcer was caused by *Staphylococcus aureus*, a common bacterium known to cause various infections. The patient sought medical help within 24 hours, allowing for prompt treatment, which led to complete healing. In contrast, the corneal ulcer shown in the Fig 2b, was presented at much late stage, one week after the onset of ulcer. Due to the delayed presentation, the infection had progressed, and the patient developed a corneal opacity. The identified bacterium in this case was *Pseudomonas aeruginosa*, an opportunistic pathogen (Fig 2a and b).

### Biochemical characterization of the bacterial isolates

The bacterial Isolates were identified and grouped into distinct clusters based on their phenotypic and biochemical characteristics. It was observed that the majority of the isolates were unable to metabolize different sugars (S2 Table). Based on the biochemical characterization strains EW1, EW2, EW4 and EW10 were identified as *Staphylococcus aureus*, of which three were methicillin resistant *Staphylococcus aureus* (MRSA). *Staphylococcus aureus* were catalase, coagulase and DNase positive, and two *staphylococci* appeared to be coagulase negative (Fig 3). Strains EW5 and EW6 were clustered together, identified as *Escherichia coli* (EC6, EC7) and utilized same sugars. Strains EW8, EW9, EW11, EW12, EW13, EW15 were grouped together in a single clade and were identified as *Pseudomonas* species. A few other bacterial isolates, including *Streptococcus* spp., (STC), *Stenotrophomonas maltophilia* (STM), as well as EC6 and EC7 demonstrated ability to utilize D-glucose, nitrate reductase. Only one isolate, *Haemophilis spp* (HAE), showed hydrogen sulphide gas production.

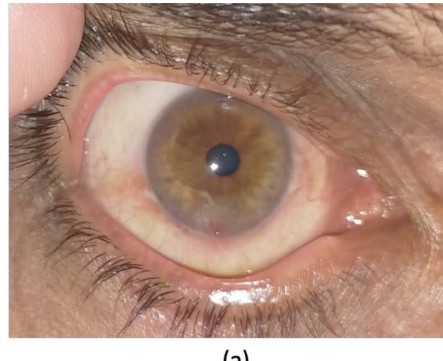
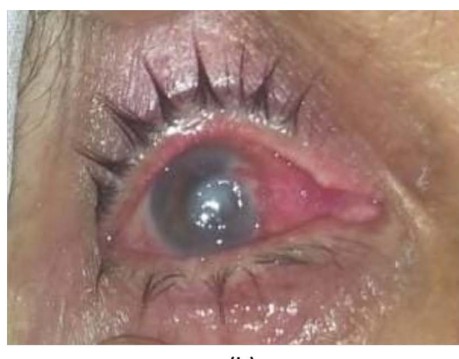

(a)          (b)

**Fig 2. Clinical outcome of the corneal ulcers (a) fully healed corneal ulcer (b) corneal opacity due to delayed treatment.**

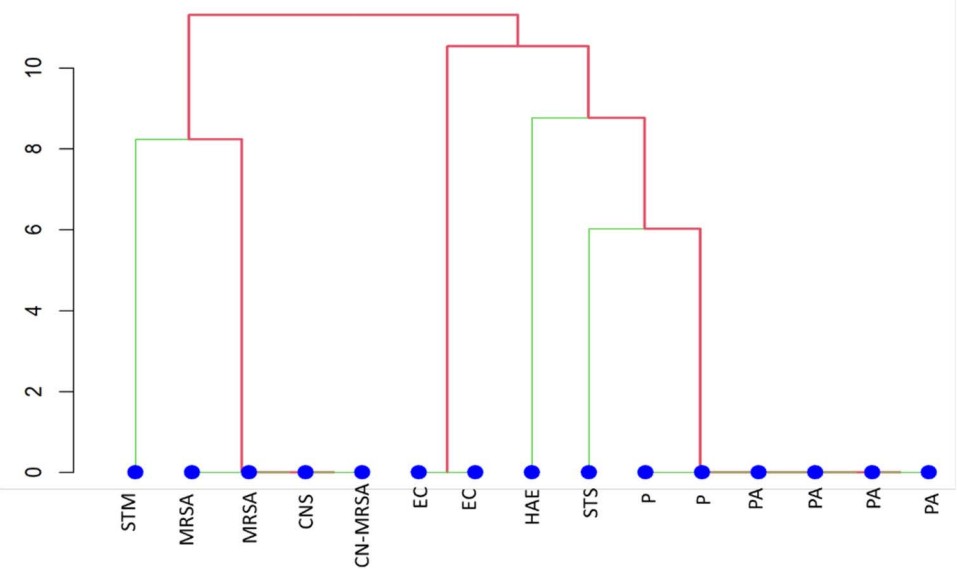

**Fig 3. Clustering of the bacterial isolates from the corneal ulcer samples was performed based on various biochemical characteristics.** Similarity between isolates was assessed through 10 biochemical sugar fermentation tests using the API 10 S strips, with results converted into binary data (0 for negative and 1 for positive) using R software. The Jacquard coefficient was employed to estimate the similarity between the isolates, and clustering was performed using the unweighted average linkage method. This approach grouped bacterial isolates with highest similarity, providing a clear visualization of their relatedness.

## Antibiotic resistance profile of bacterial species

Methicillin-resistant coagulase-negative *staphylococci* (MR-CoNS) exhibited a multidrug-resistant profile, showing resistance to ceftriaxone (zone <24 mm), cloxacillin (zone ≤10 mm), co-trimoxazole (zone ≤10 mm), imipenem (zone ≤16 mm), cephradine (zone ≤14 mm), augmentin, and ampicillin (zone ≤10 mm). This resistance pattern is consistent with typical MR-CoNS phenotypes, where *mecA*-mediated β-lactam resistance frequently coexists with resistance to other antibiotic classes due to selective pressures in healthcare settings [33,34]. Studies from India have reported widespread dissemination of the *mecA* gene, contributing to the rise of MR-CoNS. Accordingly, this study sought to assess the prevalence of *mecA* and its correlation with antimicrobial resistance [33,35]. Vancomycin susceptibility testing revealed MICs of ≤2 µg/mL for the MRSA isolates, classifying them as sensitive according to CLSI guidelines. *Streptococcus* spp. isolates were resistant to ceftriaxone and penicillin (both zones <24 mm) but remained susceptible to linezolid (zone ≥21 mm). This resistance may be attributed to mutations in penicillin-binding proteins (PBPs) or efflux mechanisms, while sustained susceptibility to linezolid likely reflects its limited clinical use and slow resistance development [36].

E. coli isolates were resistant to ceftriaxone (zone <23 mm), co-trimoxazole (zone ≤10 mm), gentamicin (zone ≤12 mm), and ampicillin (zone ≤13 mm) but remained susceptible to amikacin (zone ≥17 mm), doxycycline (zone ≥14 mm), and imipenem (zone ≥23 mm). This profile is consistent with extended-spectrum β-lactamase (ESBL)-producing strains [37]. Among *Pseudomonas* isolates (n = 6), one strain exhibited resistance to both amikacin (zone ≤14 mm) and gentamicin (zone ≤12 mm), while all isolates were uniformly resistant to ciprofloxacin (zone ≤15 mm). This resistance pattern suggests the presence of upregulated efflux pumps conferring fluoroquinolone resistance, along with sporadic occurrence of aminoglycoside-modifying enzymes [38].

## Species-specific antibiotic resistance trends

The distribution of resistance levels across various antibiotics for different bacterial species can be seen in Fig 4. The antibiotic ciprofloxacin shows the highest cumulative resistance level, with a notable contribution from various bacterial species. Methicillin-resistant, coagulase-negative *staphylococci* (MR-CoNS) and EC6 show significant resistance, indicating these bacteria are particularly resistant to ciprofloxacin. This suggests ciprofloxacin is less effective against these species. Both gentamicin and co-trimoxazole show high cumulative resistance levels. For gentamicin, a significant portion of resistance comes from methicillin resistant *Staphylococcus aureus* (MRSA) strains. Co-trimoxazole also shows high resistance levels, primarily contributed by EC6 and MR-CoNS. This indicates these antibiotics face substantial resistance from multiple bacterial species, suggesting limited effectiveness. Whereas amikacin shows lower resistance levels compared to ciprofloxacin and gentamicin. The resistance is primarily from methicillin resistant *Staphylococcus aureus* strains, with minor contributions from other species. This suggests amikacin is more effective against a broader range of bacteria. Vancomycin shows relatively overall lower resistance levels, suggesting that vancomycin remains effective against many bacterial species, except MRSA still poses a challenge. Overall, ciprofloxacin, gentamicin, and co-trimoxazole exhibit high resistance levels from multiple bacterial species, indicating the need for alternative treatments or combination therapies

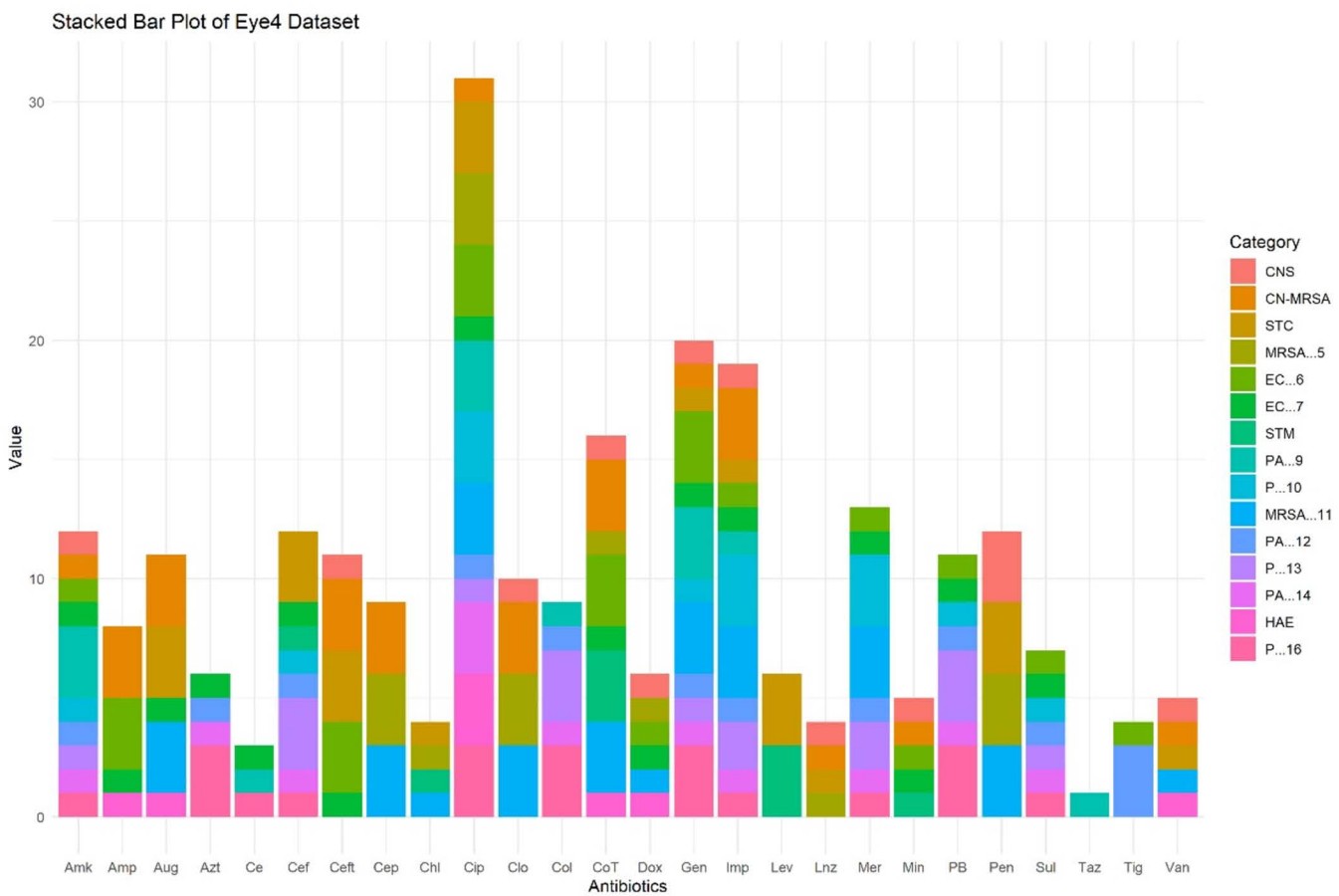

**Fig 4. The stacked bar plot presents the distribution of resistance levels across various antibiotics for different bacterial species.** The x-axis represents the antibiotics, while the y-axis indicates the cumulative resistance levels. Each color in the bars corresponds to a specific bacterial species, as denoted in the legend on the right.

to manage infections caused by these resistant strains. Antibiotics like vancomycin and amikacin show relatively lower resistance levels, suggesting they are more effective against a broader range of bacterial species. The high resistance observed in MRSA strains across multiple antibiotics highlights the need for targeted interventions and continuous monitoring to manage and mitigate antibiotic resistance in these species.

The distribution of resistance levels among various bacteria is shown in Fig 5. Each bacterial species is represented by a distinct color, allowing for a clear comparison of their resistance profiles. MR-CoNS (shown in pink) demonstrates a notable density at resistance level 3, signifying a significant ($P<0.05$) proportion of this bacteria type being resistant to antibiotics. This high level of resistance is concerning and underscores the need for targeted interventions to manage and mitigate resistance in MR-CoNS. Coagulase-negative *Staphylococci* (depicted in orange) peaks around level 1, indicating that these bacteria are generally sensitive to antibiotics. This suggests that standard antibiotic treatments remain effective for Coagulase-negative *Staphylococci*. Both EC6 and EC7 (represented in shades of yellow and light green, respectively) exhibit a bimodal distribution with peak at 1 and 3, indicating substantial variability in resistance levels. This suggests the presence of both low and high resistance strains within these types, complicating treatment efforts and highlighting the need for personalized treatment strategies based on specific resistance profiles [39]. The significant ($P<0.05$) variability suggests the need for personalized treatment strategies based on specific resistance profiles. This implies sensitivity to the antibiotics used.

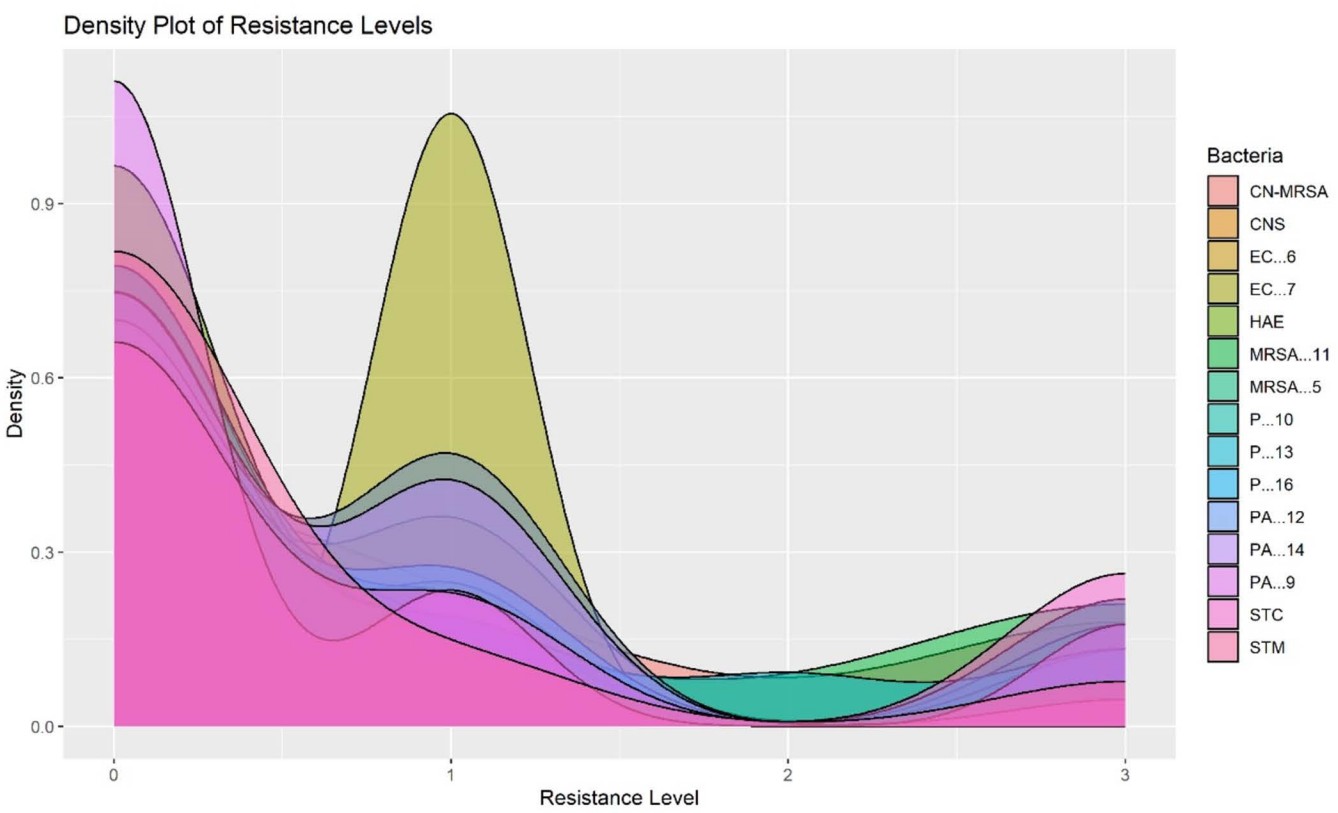

**Fig 5. Each color represents a different bacterial type, showing how resistance levels are spread for each.** The y-axis indicates the density, and the x-axis represents the resistance levels. The density plot illustrates the distribution of resistance levels for various bacterial species, classified into four categories: 0 (antibiotic not used), 1 (sensitive), 2 (intermediate), and 3 (resistant).

MRSA11 and MRSA5 (depicted in shades of blue) display significant ($P<0.05$) peaks at resistance level 3, similar to MR-CoNS. MRSA strains showed resistance to imipenem, co-trimoxazole, cloxacillin, cephradine, ceftriaxone, augmentin, ampicillin, cloxacillin, penicillin, ciprofloxacin, gentamicin, meropenem. This highlights the high resistance in these MRSA strains, pointing towards a critical need for alternative therapeutic strategies and resistance management practices [40]. The *P. aeruginosa* (PA) groups, represented in various shades of purple) show high resistance to some commonly used antibiotics such as (gentamicin, amikacin, ciprofloxacin, polymyxin B, colistin, aztreonam, ceftazidime, imipenem, meropenem, tigecycline), suggesting continuous monitoring [41]. Lastly, *Streptococcus* spp., (STC) and *Stenotrophomonas maltophilia* (STM) primarily peak at level 1, indicating that these bacteria are generally sensitive to antibiotics, which is a positive indication for current treatment efficacy. This density plot underscores the diverse responses of bacterial species to antibiotics, with particular concern for those exhibiting high resistance levels like MRSA and *P. aeruginosa* strains. The variability in antibiotic sensitivity and resistance highlights the necessity for continuous surveillance, prudent antibiotic use, and the development of new treatment strategies to effectively manage and reduce antibiotic resistance in these bacterial populations [42]. The wide distribution of resistance levels among bacteria in clinical settings underscores the challenges in treating infections caused by these bacteria.

## Comparative resistance trends

Our study documented concerning resistance patterns to empiric antibiotics for corneal ulcers, revealing both regional consistencies and global divergences. *P. aeruginosa* isolates demonstrated particularly alarming resistance to ciprofloxacin (85%), substantially exceeding global averages of 50–65% [41] while matching South Asian reports of 75–90% resistance from India [25,26,43] and 80–90% from Pakistan [24]. This regional epidemic of fluoroquinolone resistance likely stems from widespread empiric use and unregulated antibiotic access [14], compelling consideration of alternative therapies like fortified aminoglycosides or combination regimens in South Asian clinical practice.

MRSA isolates mirrored this troubling pattern, exhibiting 65–85% ciprofloxacin resistance comparable to South Asian reports [26,44] and global averages of 50–70% [41,45,46], further implicating Pakistan's fluoroquinolone misuse [14] in undermining this drug's empirical utility. While gentamicin resistance in MRSA (67%) surpassed global rates (50–60%), amikacin retained efficacy against *E. coli* vs. studies showed about 15–25% resistance regionally [2,47]. This suggests amikacin's potential as a first-line alternative in settings with high gentamicin resistance, though local surveillance is critical given variability in *P. aeruginosa* susceptibility (33% resistant to gentamicin in our study.

Carbapenem resistance in *P. aeruginosa* (40%) was double global rates (10–20%) [48,49], reflecting regional dissemination of metallo-β-lactamase genes. Conversely, vancomycin resistance in MRSA remained low, consistent with Pakistan's restricted use [23,47], affirming its role for Gram-positive infections despite rising VRSA reports elsewhere. These findings collectively mandate urgent implementation of antimicrobial stewardship programs, enhanced diagnostic capacity, and regionally tailored treatment guidelines to address this mounting public health crisis.

## Frequency of interactions between each antibiotic and bacterial strain

The frequency of interactions between each antibiotic and bacterial strain is visualized using chord diagram (Fig 6). The significant variability in resistance levels across antibiotics underscores the challenge of treating bacterial infections effectively. Some antibiotics face widespread resistance, making them less reliable, while others still show potential effectiveness [50]. Each antibiotic and bacterial strain is represented by a segment along the circumference of the circle. The connections (chords) between segments represent the resistance levels. The width of each chord indicates the resistance level, with thicker chords representing higher resistance levels and thinner chords representing lower resistance levels [51]. Antibiotics like ciprofloxacin and co-trimoxazole have multiple thick connections to various bacterial strains, indicating high resistance levels [48,52]. Conversely, antibiotics like linezolid and meropenem have fewer and thinner chords, indicating lower resistance levels. Amikacin shows interactions with several bacterial strains, indicating its broad usage. ceftriaxone also shows

multiple interactions, suggesting its extensive use across different strains. Bacterial Strains including Coagulase-Negative *Staphylococci* and methicillin-resistant Coagulase-Negative *staphylococci* (MR-CoNS) show interactions with multiple antibiotics, indicating a pattern of multi-drug usage possibly due to resistance issues. Coagulase-Negative *Staphylococci's*

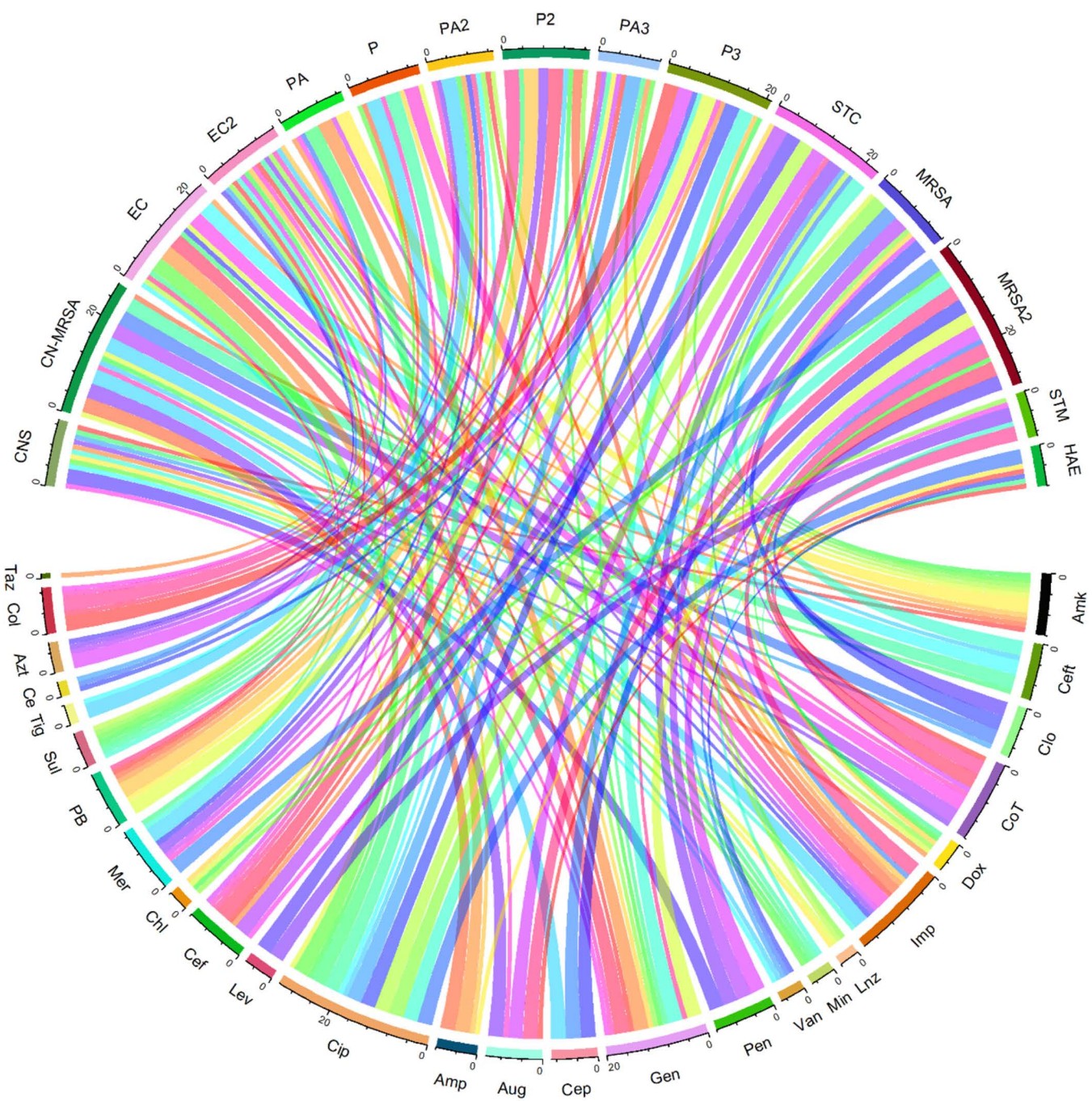

**Fig 6. The chord diagram representing the interactions between various antibiotics and bacterial strains, providing a visual summary of their relationships, created using Circlize package in R Studio [ 53].**

interactions with amikacin, ceftriaxone, and clindamycin highlight its multi-drug resistance profile, which is a significant concern in clinical settings. *Streptococcus* spp., and MRSA also show significant interactions, suggesting challenges in treating these strains effectively. By understanding these interactions, healthcare providers can make informed decisions to optimize treatment protocols and mitigate the development of antibiotic resistance.

The relationships between various bacterial categories based on their antibiotic resistance profiles are shown in correlation matrix plot (Fig 7). Strong positive correlations, indicated by dark blue circles, are observed within certain groups, such as between MRSA categories such as MRSA11 and MRSA5 and within the *P. aeruginosa* categories. These strong correlations suggest that these bacteria might share similar resistance mechanisms, possibly due to genetic similarities or horizontal gene transfer. Conversely, negative correlations, indicated by lighter red circles, are less frequent but present, such as between *P. aeruginosa* and *S. aureus* strains, suggesting inverse resistance patterns possibly due to competitive interactions or different ecological niches [54].

## Conclusion

The emergence and spread of antibiotic resistance represent a critical global health challenge, contributing significantly to increased morbidity and mortality across all regions. This study characterized antibiotic resistance patterns in a select group of clinical isolates, revealing key trends that warrant further investigation in larger cohorts. Most notably, we

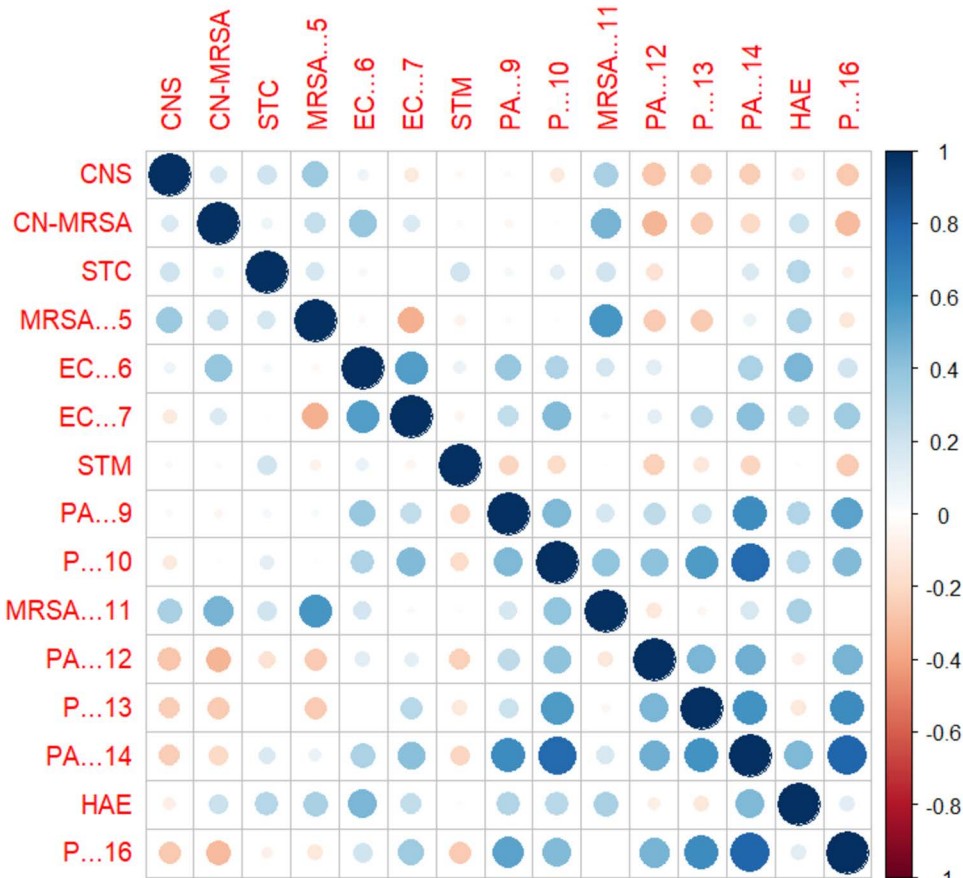

**Fig 7. The correlation matrix plot displays the relationships between various bacterial categories based on their antibiotic resistance profiles.** The axes of the plot represent the same bacterial categories, and the color and size of the circles denote the strength and direction of the correlations.

observed complete resistance to ciprofloxacin across all Pseudomonas isolates and high-level multidrug resistance in MR-CoNS strains—findings consistent with established regional resistance patterns. These results align with surveillance data suggesting declining fluoroquinolone efficacy against Gram-negative pathogens and persistent β-lactam resistance in staphylococci. Understanding these resistance dynamics is essential for guiding clinical decision-making and implementing effective infection control measures. While these preliminary findings demonstrate the value of local susceptibility monitoring for detecting emerging trends, future studies with expanded isolate collection and molecular characterization of resistance mechanisms would strengthen the evidence base.

## Supporting information

**S1 Table. Antibiotic resistant spectra of bacterial isolates from corneal ulcer samples.**
(DOCX)

**S2 Table. Biochemical characters of the bacterial isolates obtained from corneal ulcer samples.**
(DOCX)

## Author contributions

**Data curation:** Sidra Abbas.

**Formal analysis:** Sidra Abbas.

**Investigation:** Naila Obaid, Hajira Younas.

**Methodology:** Naila Obaid.

**Project administration:** Naila Obaid.

**Software:** Sidra Abbas.

**Validation:** Ayesha Saeed, Shaghufta Perveen.

**Visualization:** Sidra Abbas, Shaghufta Perveen.

**Writing – original draft:** Sidra Abbas.

**Writing – review & editing:** Sidra Abbas, Ayesha Saeed, Shaghufta Perveen.

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
