## [Decision Letter · Decision Letter 0]

19 Jan 2025

Dear Dr. Abbas,

We look forward to receiving your revised manuscript.

Kind regards,

Mengistu Hailemariam Zenebe, PhD

Academic Editor

PLOS ONE

Journal Requirements:

4. We note that Figure 1 in your submission contain map images which may be copyrighted. All PLOS content is published under the Creative Commons Attribution License (CC BY 4.0), which means that the manuscript, images, and Supporting Information files will be freely available online, and any third party is permitted to access, download, copy, distribute, and use these materials in any way, even commercially, with proper attribution. For these reasons, we cannot publish previously copyrighted maps or satellite images created using proprietary data, such as Google software (Google Maps, Street View, and Earth). For more information, see our copyright guidelines: http://journals.plos.org/plosone/s/licenses-and-copyright.

1)  You may seek permission from the original copyright holder of Figure 1 to publish the content specifically under the CC BY 4.0 license.  

2) If you are unable to obtain permission from the original copyright holder to publish these figures under the CC BY 4.0 license or if the copyright holder’s requirements are incompatible with the CC BY 4.0 license, please either i) remove the figure or ii) supply a replacement figure that complies with the CC BY 4.0 license. Please check copyright information on all replacement figures and update the figure caption with source information. If applicable, please specify in the figure caption text when a figure is similar but not identical to the original image and is therefore for illustrative purposes only.

Reviewers' comments:

Reviewer's Responses to Questions

**Comments to the Author**

1. Is the manuscript technically sound, and do the data support the conclusions?

Reviewer #1: Yes

Reviewer #2: Yes

2. Has the statistical analysis been performed appropriately and rigorously?

Reviewer #1: Yes

Reviewer #2: Yes

3. Have the authors made all data underlying the findings in their manuscript fully available?

Reviewer #1: Yes

Reviewer #2: Yes

4. Is the manuscript presented in an intelligible fashion and written in standard English?

Reviewer #1: Yes

Reviewer #2: Yes

Reviewer #1: The article presents significant findings on antibiotic resistance patterns from corneal ulcers. Globally, antibiotic resistance is recognized as a major public health threat. These infections can lead to visual impairment and even blindness if not treated. Pseudomonas aeruginosa shows high resistance to commonly used antibiotics such as ciprofloxacin and gentamicin, which are frequently prescribed for corneal infections. This trend is alarming, necessitating that the protocols may need to be reevaluated to manage these infections.

In the context of Pakistan, the study's implications are particularly pronounced. The country faces a significant burden of infectious diseases, compounded by inadequate healthcare infrastructure and the overuse of antibiotics. Even though the sample size is small, I have not seen such a study reported in the Pakistani population. There is a lack of data on antibiotic resistance patterns in Pakistan concerning corneal ulcers. The current study fills this gap by providing valuable insights. The appealing visual illustrations of resistance patterns enhance the understanding of the relationships between different antibiotics and bacterial strains. This study not only informs clinical practice but also underscores the urgent need for public health initiatives to address antibiotic misuse.

Overall, I suggest minor revisions to improve the quality of the manuscript further.

Introduction

Explain antibiotics use and the prevalence of resistance in Pakistan, especially in ophthalmic infections?

Highlight how local practices may contribute to resistance patterns?

Methodology

Please elaborate on the disc diffusion method. Discuss how the results were interpreted, including the criteria for determining resistance and susceptibility. This is crucial for readers to understand the significance of your findings.

Discussion

It is important to acknowledge the role of non-bacterial agents in corneal infections. Discuss the prevalence of fungal pathogens such as Aspergillus and Candida, as well as viral and parasitic causes.

Highlight how the resistance patterns observed in the current study reflect or diverge from global trends.

Reviewer #2: The study's measurement of healing rates in corneal ulcer patients, alongside assessing antibiotic resistance levels across various bacterial strains and their specific treatment responses, is highly relevant in the context of rising antibiotic resistance. Understanding the healing rates allows for the evaluation of treatment efficacy while analyzing resistance patterns, which provides critical insights into the challenges faced in managing corneal infections as well as informs more personalized treatment plans towards optimizing patient outcomes. However, here are some comments and recommendations.

• The timeline of the study should be reviewed and consistently stated throughout the document to ensure clarity.

• The sample size utilized in this study is insufficient.

• Currently, relying solely on disc diffusion test results is not widely accepted for detecting antibiotic resistance. Therefore, it is recommended that the authors validate their disc diffusion results, particularly for resistant strains, using additional approved methods such as the microdilution method or E-test. Employing these techniques will provide accurate minimum inhibitory concentration (MIC) values for each resistant antibiotic strain identified.

• The detection of vancomycin resistance through disc diffusion is not considered valid according to CLSI and EUCAST guidelines. It is essential to perform E-tests alongside microdilution methods to assess vancomycin resistance and investigate the presence of VanA, VanB, and VanC genes in the isolated strains.

• Although phenotypic approaches are valuable, integrating high-throughput molecular methods would yield data, leading to stronger conclusions. Notwithstanding, the authors did impressive work regarding their statistical analysis. The highlights of some of the statistical analysis, especially from the frequency of interactions between each antibiotic and bacterial strain, can be stated in the abstract.

**Do you want your identity to be public for this peer review?** For information about this choice, including consent withdrawal, please see our Privacy Policy

Reviewer #1: No

Reviewer #2: **Yes: ** ALEXANDER KWARTENG

---

## [Author Response · Author response to Decision Letter 1]

17 Feb 2025

Response to Reviewers’ Comments

The authors are thankful to the reviewers for their valuable insights and constructive feedback which has significantly contributed to enhance the quality of the manuscript. All the revisions are highlighted in yellow.

Reviewer # 1

General comment

The article presents significant findings on antibiotic resistance patterns from corneal ulcers. Globally, antibiotic resistance is recognized as a major public health threat. These infections can lead to visual impairment and even blindness if not treated. Pseudomonas aeruginosa shows high resistance to commonly used antibiotics such as ciprofloxacin and gentamicin, which are frequently prescribed for corneal infections. This trend is alarming, necessitating that the protocols may need to be reevaluated to manage these infections. In the context of Pakistan, the study's implications are particularly pronounced. The country faces a significant burden of infectious diseases, compounded by inadequate healthcare infrastructure and the overuse of antibiotics. Even though the sample size is small, I have not seen such a study reported in the Pakistani population. There is a lack of data on antibiotic resistance patterns in Pakistan concerning corneal ulcers. The current study fills this gap by providing valuable insights. The appealing visual illustrations of resistance patterns enhance the understanding of the relationships between different antibiotics and bacterial strains. This study not only informs clinical practice but also underscores the urgent need for public health initiatives to address antibiotic misuse.

Overall, I suggest minor revisions to improve the quality of the manuscript further.

Comment 1

Explain antibiotics use and the prevalence of resistance in Pakistan, especially in ophthalmic infections?

Response

The required information regarding antibiotics use and the prevalence of resistance in Pakistan, particularly in ophthalmic infections, has been added in lines 78-80.

Comment 2

Highlight how local practices may contribute to resistance patterns?

Response

The required information on how local practices contribute to resistance patterns has been incorporated from line 76. This section now includes details on antibiotic overuse, self-medication, and other key factors influencing resistance trends.

Comment 3

Please elaborate on the disc diffusion method. Discuss how the results were interpreted, including the criteria for determining resistance and susceptibility. This is crucial for readers to understand the significance of your findings.

Response 3

The disc diffusion method, also known as the Kirby-Bauer method, is a widely used technique for assessing the antibiotic susceptibility of bacterial isolates. This method follows the guidelines set by the Clinical and Laboratory Standards Institute (CLSI). The description is added from line 126.

Comment 4

It is important to acknowledge the role of non-bacterial agents in corneal infections. Discuss the prevalence of fungal pathogens such as Aspergillus and Candida, as well as viral and parasitic causes.

Response

The introduction has been expanded from line 61 to include a discussion on the prevalence of fungal pathogens, such as Aspergillus and Candida, as well as viral and parasitic causes of corneal infections. Relevant sources have been cited to highlight the significance of these pathogens in corneal infections.

Reviewer # 2

General comment

Highlight how the resistance patterns observed in the current study reflect or diverge from global trends.

The study's measurement of healing rates in corneal ulcer patients, alongside assessing antibiotic resistance levels across various bacterial strains and their specific treatment responses, is highly relevant in the context of rising antibiotic resistance. Understanding the healing rates allows for the evaluation of treatment efficacy while analysing resistance patterns, which provides critical insights into the challenges faced in managing corneal infections as well as informs more personalized treatment plans towards optimizing patient outcomes. However, here are some comments and recommendations.

Comment 1

The timeline of the study should be reviewed and consistently stated throughout the document to ensure clarity.

Response

The timeline has been carefully reviewed and ensure consistency throughout the document to improve clarity.

Comment 2

The sample size utilized in this study is insufficient.

Response

The incidence of corneal infections is relatively low, and patient recruitment was inherently limited by case availability during the study period. Notably, corneal ulcer cases are more prevalent during the summer and monsoon seasons, which may be attributed to increased exposure to environmental pathogens and humid conditions favouring microbial growth. Nevertheless, the findings provide valuable insights into corneal ulcer pathology and microbial patterns. Future studies with larger cohorts will further substantiate these observations.

Comment 3

Currently, relying solely on disc diffusion test results is not widely accepted for detecting antibiotic resistance. Therefore, it is recommended that the authors validate their disc diffusion results, particularly for resistant strains, using additional approved methods such as the microdilution method or E-test. Employing these techniques will provide accurate minimum inhibitory concentration (MIC) values for each resistant antibiotic strain identified.

Response

Due to economic constraints in Pakistan, the disc diffusion method remains the most widely used and feasible approach in clinical settings. Despite its limitations, it provides reliable and reproducible results under local laboratory conditions. In the current study, E-test is typically performed for vancomycin-resistant strains; however, as no vancomycin resistance was observed, further MIC determination was not required.

Comment 4

The detection of vancomycin resistance through disc diffusion is not considered valid according to CLSI and EUCAST guidelines. It is essential to perform E-tests alongside microdilution methods to assess vancomycin resistance and investigate the presence of VanA, VanB, and VanC genes in the isolated strains.

Response

No vancomycin-resistant strains were identified in this study. The authors acknowledges that as per CLSI and EUCAST guidelines, disc diffusion alone is not a reliable method for detecting vancomycin resistance. However, all isolates showed susceptibility to vancomycin based on the disc diffusion method, and no resistance patterns were observed that would justify further confirmation by E-test, broth microdilution, or PCR for VanA, VanB, and VanC genes. Future studies will consider molecular confirmation where necessary.

Comment 5

Although phenotypic approaches are valuable, integrating high-throughput molecular methods would yield data, leading to stronger conclusions. Notwithstanding, the authors did impressive work regarding their statistical analysis. The highlights of some of the statistical analysis, especially from the frequency of interactions between each antibiotic and bacterial strain, can be stated in the abstract.

Response

Abstract has been modified accordingly.

I hope that the responses have satisfied the queries/ comments of the reviewers. Once again I am thankful to the reviewers whose input has raised the quality of the manuscript.

Dr. Sidra Abbas

---

## [Decision Letter · Decision Letter 1]

16 Mar 2025

Thank you for submitting your manuscript to PLOS ONE. After careful consideration, we feel that it has merit but does not fully meet PLOS ONE’s publication criteria as it currently stands. Therefore, we invite you to submit a revised version of the manuscript that addresses the points raised during the review process.

We look forward to receiving your revised manuscript.

Kind regards,

Mengistu Hailemariam Zenebe, PhD

Academic Editor

PLOS ONE

Journal Requirements:

Reviewers' comments:

Reviewer's Responses to Questions

**Comments to the Author**

Reviewer #1: (No Response)

Reviewer #2: All comments have been addressed

2. Is the manuscript technically sound, and do the data support the conclusions?

Reviewer #1: Partly

Reviewer #2: Yes

3. Has the statistical analysis been performed appropriately and rigorously?

Reviewer #1: Yes

Reviewer #2: Yes

4. Have the authors made all data underlying the findings in their manuscript fully available?

Reviewer #1: Yes

Reviewer #2: Yes

5. Is the manuscript presented in an intelligible fashion and written in standard English?

Reviewer #1: Yes

Reviewer #2: Yes

Reviewer #1: The study addresses a critical public health issue of antibiotic resistance in corneal ulcer pathogens in Pakistani region with limited data. The topic is timely and clinically relevant as I stated in my previous review comments also. But I have noticed many discrepancies in the revised manuscript which are as following;

In response to the comment “Explain antibiotics use and the prevalence of resistance in Pakistan, especially in ophthalmic infections”, The authors addressed the comment partially which is not satisfactory. The authors acknowledged empirical antibiotic use (fluoroquinolones, aminoglycosides) and drivers of resistance (self-medication, over the counter availability of drugs). Quantitative data on resistance prevalence, such as national and regional resistance from Pakistan, is missing. The text lacks data and citations on local ophthalmic infections related antibiotic resistance.

The authors response to address the comment “Please elaborate on the disc diffusion method. Discuss how the results were interpreted, including the criteria for determining resistance and susceptibility. This is crucial for readers to understand the significance of your findings.” is incomplete. The CLSI reference is noted, but critical details are missing. Specific CLSI breakpoints for each antibiotic have not been enlisted. No quality control strains have been used to validate the method.

I cannot find the addressal of the comment “Highlight how the resistance patterns observed in the current study reflect or diverge from global trends.” In the discussion section of the article. The authors fail to compare findings to regional studies. For example, there is no mention of P. aeruginosa resistance rates to ciprofloxacin in South Asia/Pakistan/India and global data. This comment has not been addressed.

The authors need to justify why fungal/viral testing was not performed despite Sabouraud agar use to confirm any co-infection in patients. Discuss the implications of missing fungal/viral co-infection data.

Kirby-Bauer disc diffusion alone for vancomycin susceptibility is invalid as per CLSI guidelines. Disc diffusion lacks reliability for glycopeptides and carbapenems.

Biochemical tests for bacterial identification lack specificity. Biochemical characterization is rudimentary. Misidentification of Staphylococcus aureus , like coagulase-negative isolates labeled as MRSA raises concerns. No secondary confirmation test was performed, such as broth microdilution and molecular assays.

It is better to confirm species using 16S rRNA sequencing.

Mixed infections are common in corneal ulcers and influence resistance profiles. Perform fungal cultures or PCR and viral testing to rule out co-infections.

The retrospective design lacks a control group, reducing the ability to infer causality or resistance trends.

The choice of 26 antibiotics is broad but lacks justification. Why Nitrofurantoin antibiotic also checked as it is used in urinary tract infections?

The conclusion is generic and lacks specificity. It overstates the study’s impact given the small sample size.

Some citations (Woodland et al., 1992, Title: Causes of conjunctivitis and keratoconjunctivitis in Karachi, Pakistan) are outdated. Include recent studies.

The authors did not mention inclusion and exclusion criteria.

Decision

Major Revisions Suggested. Most of the comments have been addressed superficially. The manuscript suffers from inconsistencies.

Reviewer #2: (No Response)

**Do you want your identity to be public for this peer review?** For information about this choice, including consent withdrawal, please see our Privacy Policy

Reviewer #1: No

Reviewer #2: No

---

## [Author Response · Author response to Decision Letter 2]

18 Apr 2025

Response to Reviewers’ Comments

The authors are thankful to the reviewers for their valuable insights and constructive feedback which has significantly contributed to enhance the quality of the manuscript. All the revisions are highlighted in yellow.

Reviewer Comments

The study addresses a critical public health issue of antibiotic resistance in corneal ulcer pathogens in Pakistani region with limited data. The topic is timely and clinically relevant as I stated in my previous review comments also. But I have noticed many discrepancies in the revised manuscript which are as following.

Comment 1

In response to the comment “Explain antibiotics use and the prevalence of resistance in Pakistan, especially in ophthalmic infections”, The authors addressed the comment partially which is not satisfactory. The authors acknowledged empirical antibiotic use (fluoroquinolones, aminoglycosides) and drivers of resistance (self-medication, over the counter availability of drugs). Quantitative data on resistance prevalence, such as national and regional resistance from Pakistan, is missing. The text lacks data and citations on local ophthalmic infections related antibiotic resistance.

Response

Quantitative data on resistance prevalence at the regional level have been added in the introduction page 3.

Comment 2

The authors response to address the comment “Please elaborate on the disc diffusion method. Discuss how the results were interpreted, including the criteria for determining resistance and susceptibility. This is crucial for readers to understand the significance of your findings.” is incomplete. The CLSI reference is noted, but critical details are missing. Specific CLSI breakpoints for each antibiotic have not been enlisted. No quality control strains have been used to validate the method.

Response

The disc diffusion method has been revised on page 5. Specific CLSI breakpoints for each antibiotic have been included results and discussion in the under the heading antibiotic resistance profile of bacterial species. However, in clinical pathology laboratory, antibiotic susceptibility testing for different samples from different departments in the hospitals including the corneal ulcers are performed as part of routine diagnostic workflow, where hundreds of samples are processed daily under standardized operating procedures. While ideal for research studies, running concurrent QC strains with every routine clinical specimen is not feasible due to:

Operational Considerations:

As a high-volume diagnostic laboratory processing hundreds of specimens daily, we prioritize, timely reporting for patient care, Batch-level QC validation, Strict adherence to standardized operating procedures

Standardized Identification Methods:

All isolates underwent biochemical confirmation using API test kits (bioMérieux), consistent with established methodologies given below as well (Khor et al., 2024; Naik et al., 2021).

Comment 3

I cannot find the addressal of the comment “Highlight how the resistance patterns observed in the current study reflect or diverge from global trends.” In the discussion section of the article. The authors fail to compare findings to regional studies. For example, there is no mention of P. aeruginosa resistance rates to ciprofloxacin in South Asia/Pakistan/India and global data. This comment has not been addressed.

Response

Resistance patterns of the tested antibiotics to regional/global data, emphasizing critical disparities have been added in the text on page 10 and 11.

Comment 4

The authors need to justify why fungal/viral testing was not performed despite Sabouraud agar use to confirm any co-infection in patients. Discuss the implications of missing fungal/viral co-infection data.

Response

Fungal and viral keratitis cases were excluded from this study, as the focus was on admitted patients with bacterial keratitis. As part of our standard microbiological protocol, corneal scrapings from all patients were cultured on Sabouraud agar to screen for potential fungal co-infections. However, patients diagnosed with fungal or viral keratitis were managed separately and not included in the final analysis to maintain a focused and uniform study population.

Comment 5

Kirby-Bauer disc diffusion alone for vancomycin susceptibility is invalid as per CLSI guidelines. Disc diffusion lacks reliability for glycopeptides and carbapenems.

Response

Broth microdilution (BMD) MIC testing has now been incorporated for vancomycin susceptibility, as required by CLSI guidelines for glycopeptides. The revised manuscript includes these updated methods in the Methodology section.

Comment 6

Biochemical tests for bacterial identification lack specificity. Biochemical characterization is rudimentary. Misidentification of Staphylococcus aureus , like coagulase-negative isolates labeled as MRSA raises concerns. No secondary confirmation test was performed, such as broth microdilution and molecular assays.

Response

We acknowledge that referring to coagulase-negative isolates as MRSA was a typographical error. In our hospital setting, biochemical identification remains a standard and practical approach for timely diagnosis and reporting, particularly given the high volume of samples and the need to initiate prompt treatment. While we recognize the superior specificity of molecular assays such as PCR, resource limitations and financial constraints precluded their use in this study. Nonetheless, the biochemical methods employed are routinely used in clinical microbiology and are considered sufficiently reliable for initial identification and management.

Comment 7

It is better to confirm species using 16S rRNA sequencing.

Response

Thank you for the suggestion. We agree that 16S rRNA sequencing is a highly reliable and accurate method for bacterial species identification. However, due to the high volume of samples and limited financial and laboratory resources, it was not feasible to incorporate molecular sequencing methods in our study. Our identification protocol relied on conventional biochemical testing, which is routinely used in clinical microbiology laboratories for timely and cost-effective diagnosis. While we acknowledge that 16S rRNA sequencing would enhance the precision of species confirmation, the focus of our study was on clinical outcomes and management, for which conventional identification methods were considered sufficient within the given constraints. Below are the few studies which used biochemical and the sensitivity assays for the identification of the bacterial strains.

Khor, W.B., Lakshminarayanan, R., Periayah, M.H., Prajna, V.N., Garg, P., Sharma, N., Mehta, J.S., Young, A., Goseyarakwong, P., Puangsricharern, V. and Tan, A.L., 2024. The antibiotic resistance profiles of Pseudomonas aeruginosa in the Asia Cornea Society Infectious Keratitis Study. International Ophthalmology, 44(1), p.361.

Fernandes, M., Vira, D., Medikonda, R. and Kumar, N., 2016. Extensively and pan-drug resistant Pseudomonas aeruginosa keratitis: clinical features, risk factors, and outcome. Graefe's Archive for Clinical and Experimental Ophthalmology, 254, pp.315-322.

Naik, P., Pandey, S., Gagan, S., Biswas, S. and Joseph, J., 2021. Virulence factors in multidrug (MDR) and Pan-drug resistant (XDR) Pseudomonas aeruginosa: a cross-sectional study of isolates recovered from ocular infections in a high-incidence setting in southern India. Journal of Ophthalmic Inflammation and Infection, 11, pp.1-11.

Cabrera‐Aguas, M., Chidi‐Egboka, N., Kandel, H. and Watson, S.L., 2024. Antimicrobial resistance in ocular infection: A review. Clinical & Experimental Ophthalmology, 52(3), pp.258-275.

AlBahrani, S., Alqazih, T.Q., Aseeri, A.A., Al Argan, R., Alkhafaji, D., Alrqyai, N.A., Alanazi, S.M., Aldakheel, D.S., Ghazwani, Q.H., Jalalah, S.S. and Alshuaibi, A.K., 2024. Pattern of cephalosporin and carbapenem-resistant Pseudomonas aeruginosa: a retrospective analysis. IJID regions, 10, pp.31-34.

Comment 8

Mixed infections are common in corneal ulcers and influence resistance profiles. Perform fungal cultures or PCR and viral testing to rule out co-infections.

Response

Thank you for the valuable comment. For all admitted patients with corneal ulcers, corneal scrapings were collected and processed for microbiological evaluation, including staining and culture on blood agar, chocolate agar, and Sabouraud dextrose agar. Smears were also prepared for microscopic examination. Patients who showed a positive culture and sensitivity to a specific antibiotic were started on targeted therapy and included in the study. Cases of viral keratitis were excluded, as these patients were managed on an outpatient basis and not part of the admitted cohort. Similarly, patients with fungal keratitis who developed complications such as non-healing ulcers or corneal perforation were excluded from the final analysis. However, patients presenting with mixed infections involving both bacterial and fungal pathogens, who responded to treatment, were included in the study.

Comment 9

The retrospective design lacks a control group, reducing the ability to infer causality or resistance trends.

Response

Thank you for the comment. However, we respectfully disagree with the suggestion regarding the need for a control group in this context. The retrospective design of our study was intentionally chosen to allow for the efficient analysis of resistance patterns in clinically confirmed cases of infectious keratitis. The primary objective was to characterize antimicrobial resistance in real-world clinical isolates rather than to assess treatment efficacy or outcomes between different groups. Given the ethical considerations, it would not be appropriate to withhold diagnostic procedures, such as corneal scrapings, from symptomatic patients solely to establish an untreated control group. Additionally, control groups are more commonly utilized in prospective clinical trials or interventional studies, whereas retrospective observational studies, particularly those focused on microbiological profiling, do not typically include control arms. Therefore, the absence of a control group does not compromise the validity of our findings within the scope of this study's objectives.

Comment 10

The choice of 26 antibiotics is broad but lacks justification. Why Nitrofurantoin antibiotic also checked as it is used in urinary tract infections?

Response

We acknowledge the reviewer’s concern regarding the inclusion of nitrofurantoin in our antibiotic panel. This was inadvertently added due to an administrative error during table compilation and was originally intended for a separate study on urinary tract infections. We sincerely apologize for this oversight and confirm that nitrofurantoin testing was not performed in our corneal ulcer study. The updated manuscript which now reflects only clinically relevant antibiotics used for ocular infections, with detailed justifications provided for each agent based on CLSI guidelines.

Comment 11

The conclusion is generic and lacks specificity. It overstates the study’s impact given the small sample size.

Response

Conclusion have been revised

Comment 12

Some citations (Woodland et al., 1992, Title: Causes of conjunctivitis and keratoconjunctivitis in Karachi, Pakistan) are outdated. Include recent studies.

Response

Recent references have been included, and the reference are updated accordingly.

Comment 13

The authors did not mention inclusion and exclusion criteria.

Response

Inclusion and exclusion criteria have now been added in the manuscript.

Decision

Major Revisions Suggested. Most of the comments have been addressed superficially. The manuscript suffers from inconsistencies.

I hope that the responses have satisfied the queries/ comments of the reviewers. Once again I am thankful to the reviewers whose input has raised the quality of the manuscript.

Dr. Sidra Abbas

---

## [Decision Letter · Decision Letter 2]

8 May 2025

Resistance trend in bacteria isolated from corneal ulcers: A retrospective analysis from Pakistan

PONE-D-24-48906R2

We’re pleased to inform you that your manuscript has been judged scientifically suitable for publication and will be formally accepted for publication once it meets all outstanding technical requirements.

Kind regards,

Mengistu Hailemariam Zenebe, PhD

Academic Editor

PLOS ONE

Additional Editor Comments (optional):

Reviewers' comments:

Reviewer's Responses to Questions

**Comments to the Author**

Reviewer #1: All comments have been addressed

2. Is the manuscript technically sound, and do the data support the conclusions?

Reviewer #1: Partly

3. Has the statistical analysis been performed appropriately and rigorously?

Reviewer #1: I Don't Know

4. Have the authors made all data underlying the findings in their manuscript fully available?

Reviewer #1: Yes

5. Is the manuscript presented in an intelligible fashion and written in standard English?

Reviewer #1: Yes

Reviewer #1: The comments have been addressed to somehow satisfactory level. The editor may proceed for acceptance or forward to another reviewer for further evaluation of the manuscript to improve its scientific rigour.

**Do you want your identity to be public for this peer review?** For information about this choice, including consent withdrawal, please see our Privacy Policy

Reviewer #1: No

---

## [Editor Report · Acceptance letter]

PONE-D-24-48906R2

PLOS ONE

Dear Dr. Abbas,

I'm pleased to inform you that your manuscript has been deemed suitable for publication in PLOS ONE. Congratulations! Your manuscript is now being handed over to our production team.

Kind regards,

on behalf of

Dr. Mengistu Hailemariam Zenebe

Academic Editor

PLOS ONE